# The Discovery and Development of Natural-Based Biomaterials with Demonstrated Wound Healing Properties: A Reliable Approach in Clinical Trials

**DOI:** 10.3390/biomedicines10092226

**Published:** 2022-09-08

**Authors:** Nur Izzah Md Fadilah, Manira Maarof, Antonella Motta, Yasuhiko Tabata, Mh Busra Fauzi

**Affiliations:** 1Centre for Tissue Engineering and Regenerative Medicine, Faculty of Medicine, Universiti Kebangsaan Malaysia, Jalan Yaacob Latiff, Bandar Tun Razak, Kuala Lumpur 56000, Malaysia; 2Department of Industrial Engineering, University of Trento, Via Sommarive 9, 38122 Trento, Italy; 3Laboratory of Biomaterials, Department of Regeneration Science and Engineering, Institute for Life and Medical Science (LiMe), Kyoto University, 53 Kawara-cho Shogoin, Sakyo-ku, Kyoto 606-8500, Japan

**Keywords:** natural products, dressings, tissue-engineered skin, biomaterial, wound healing, clinical trial

## Abstract

Current research across the globe still focuses strongly on naturally derived biomaterials in various fields, particularly wound care. There is a need for more effective therapies that will address the physiological deficiencies underlying chronic wound treatment. The use of moist bioactive scaffolds has significantly increased healing rates compared to local and traditional treatments. However, failure to heal or prolonging the wound healing process results in increased financial and social stress imposed on health institutions, caregivers, patients, and their families. The urgent need to identify practical, safe, and cost-effective wound healing scaffolding from natural-based biomaterials that can be introduced into clinical practice is unequivocal. Naturally derived products have long been used in wound healing; however, clinical trial evaluations of these therapies are still in their infancy. Additionally, further well-designed clinical trials are necessary to confirm the efficacy and safety of natural-based biomaterials in treating wounds. Thus, the focus of this review is to describe the current insight, the latest discoveries in selected natural-based wound healing implant products, the possible action mechanisms, and an approach to clinical studies. We explore several tested products undergoing clinical trials as a novel approach to counteract the debilitating effects of impaired wound healing.

## 1. Introduction

The global number of skin damage and injury cases has significant healthcare implications and accounts for about half of the world’s annual spending in the healthcare sector. Understandably, healing treatments that aim to heal skin wounds have a long-cited history [1]. Such treatments are characterized by many aspects, ranging from long-term pressure to the four distinct healing phases of skin ulcers, including inflammation, proliferation, migration, and remodeling [2,3]. There are many types of wounds, classified as acute incision or excision wounds, that go through a regular healing process. Regrettably, chronic wounds possess aberrant healing conditions [4]. The wound healing process is strongly regulated by the secretion of various growth factors, key cytokines, and chemokines [5,6,7]. Chronic wound formation is initiated by the disruption of cellular and molecular signaling during these stages. Early excision and autografting are still the current standards for the surgical management of full-thickness wounds to avoid delayed wound healing. In the last two decades, the construction of skin tissue engineering from natural-based products with the advent of newer fabrication strategies has shown promise in treating various skin-related disorders, such as burns and deep wounds [8,9,10]. Natural-based biomaterials are incorporated into products that can reduce evaporative water loss and exudation of protein-rich fluids, prevent wound drying, and inhibit microbial reproduction [11]. Worldwide best practice guidelines for treating chronic wounds suggest redressing and applying infection control measures, and these are critical factors in the limited coherent clinical evidence for many approved products for active wound areas.

Wound care management varies according to wound categorization, tissue types and characteristics, intrinsic regenerative ability, and other environmental factors [12,13,14]. The treatment strategies and their comparative effectiveness in wound healing, specifically, in the occlusion of injured tissue, can strongly depend on the materials used in the wound dressing. Wound healing therapies using different biomaterials have been researched experimentally, and a plethora of information regarding the role of natural-based wound dressings in alleviating the causes of delayed wound healing can be found in several studies [15,16,17]. Previously, recent advancements in biomaterial-based regenerative strategies that augment the skin tissue wound healing process have been reviewed. The authors have discussed the designing of nanoengineered biomaterials, which are gaining significant attention due to their numerous functionalities for triggering wound repair [18]. To date, the advancement of extracellular matrix (ECM)-based biomaterials with various technologies for fabrication has been presented [19]. For this reason, there is a need to develop readily available, natural-based products, and cost-effective skin substitutes with features for clinical application in full-thickness skin defects [20]. It is worth noting that modern biomaterial-based wound dressings can integrate multiple functions, such as maintaining a moist environment, managing exudates, antibacterial capacity, injectability, and suitable mechanical properties, in more complicated situations [21,22].

According to the current scientific literature, there is no previous study that has evaluated, compared, and discussed the specific roles of natural-based products incorporated into biomaterials or their impact towards improving the treatment of nonhealing wounds (in vitro, in vivo, and/or clinical). This review focuses on the concept of wound healing approaches in skin tissue engineering, including the development of bioactive tissue scaffolds. The aim of this review article is to broadly summarize recent discoveries of natural-based products with great potential for wound healing and skin tissue engineering, together with their implications for clinical trials (human research). Considering the current achievements and clinical needs, there are two main issues that need to be carefully evaluated when fabricating the scaffolds for use in wound healing. First is the selection of the matrix biomaterial; biocompatibility and biodegradability are two critical issues affecting the biomedical application of synthetic and natural polymers [23,24]. Second is the structural morphology of the biomaterials, which is notably determined by their properties, performance, and fabrication method [25]. In general, two-dimensional (2D) scaffolds, such as membrane [26], film [27], and fiber [28] scaffolds, exhibit strong resistance to water, high oxygen permeability, and tough mechanical properties. In contrast, three-dimensional (3D) networks with porous structures, including foams [29], sponges [30], and hydrogels [31,32,33], can maintain a moist environment, absorb large amounts of exudate, and act as carriers for cells and bioactive substances [34]. These biomaterials can be applied depending on their morphology. Additionally, Zhong et al. previously summarized the characteristics of antibacterial hydrogels originating from natural polymers used as wound dressings for infected wound treatment [35]. Typically, 2D biomaterials are used as candidates for wound dressing, while biomaterials with 3D structures can be designated as wound dressings and bioactive tissue scaffolds.

## 2. Wound Healing

Wound repair is a normal and complex biological dynamic process occurring in all tissues and organs in the human body. It depends on the underlying disease, type of injury, systemic mediators, and local wound factors [36]. The healing of skin wounds is determined by four overlapping phases: hemostasis, inflammation, proliferation, and remodeling. These dynamic processes involve interactions between the epidermal and dermal cells, regulated angiogenesis, ECM, and plasma-derived proteins (adjusted by growth factors and cytokines) [3]. During the hemostasis phase, bleeding is controlled by sympathetic vasoconstriction followed by clot formation [37]. Various types of immune cells from the blood vessels are attracted to the site of injury and secrete pro-inflammatory cytokines. At the time of injury up to a couple of days after, migrating keratinocytes, fibroblasts, and endothelial cells from the wound edge start to secrete various growth factors as the initial step in the repair process. Subsequently, an epithelial layer is formed to cover the wound surface, coinciding with granulation tissue growth to close the wound surface. The growth of newly formed granulation tissue involves fibroblasts proliferation, collagen and ECM deposition, and new blood vessel (angiogenesis) development. At this stage, the wound will be contracted, and the wound size will be reduced due to the collagen being synthesized. Throughout the process, type III collagen (a typical constituent in the granulation tissue) is replaced by type I collagen (a major constituent in the normal human dermis). The remodeling process begins 2–3 weeks after injury, and continues for up to 2 years or more. Accordingly, the structural integrity and functional competence of the tissue will be restored [38,39]. However, the composition and structure of the skin differs between individuals, as does the rate of healing [40]. The wound healing stages and the sequential changes in the environment are depicted in Figure 1.

The impairment of wound healing is caused by many factors, and specific biological markers characterize such impairment in chronic wounds. The factors can be classified as local or systemic, and both contribute towards delayed wound healing. Local factors are referred to as foreign bodies, tissue maceration, biofilm, wound infection, ischemia, and hypoxia. Meanwhile, systemic factors include malnutrition, chronic organ diseases, diabetes, and advanced age [41]. With this in mind, it is important that we discover and develop natural-based biomaterials, applying good clinical practice, that are capable of reducing the impact of these factors. The relationships between evaluation parameters with advanced functions of wound dressings, such as adhesion, antimicrobial properties, antioxidant, anti-inflammatory, stimulus response, self-healing, conductivity, and wound monitoring features, are specifically discussed. The authors also describe the different applications of wound dressings for the different types of wounds; for example, excisional and incisional wounds [16]. More research needs to be conducted to significantly support the translatability of these findings into experimental models of clinical human wound scenarios. Current research on the treatment of chronic wounds is encouraged to focus on understanding the underlying molecular mechanisms in clinically different wounds and at various stages of healing progression and development. As mentioned above, the involvement of immune cells in wound healing has long been suspected to control and complicate the occurrence of cellular and biochemical events designed as to restore tissue integrity after injury [42].

### 2.1. Immune Cells in Wound Healing

The immune system plays an integral and critical role throughout the phases of successful wound healing. Nevertheless, it has been shown that their role and purposes are different during the healing process. The immune system is referred to as the cellular defense mechanism, as consists of innate immune cells, namely neutrophils, macrophages, and lymphocytes. These factors are the key regulators and players in wound healing [43].

Neutrophils are the first immune cells to arrive at the wound site, appearing about 24 h after injury. They are often considered the first line of defense against infection because their numbers peak at the wound site soon after injury, presumably to decontaminate the wound of foreign debris and defend against possible infections [44]. However, it was later found that the extended presence of neutrophils is characteristic of nonhealing wounds. One study demonstrated that wound healing speeds up when the neutrophil concentration is reduced at the time of wound induction but restored after 5 days [45]. Notably, increased age will prolong wound healing progression. For instance, an animal study showed that wound closure in young mice about 2 months of age occurs independently of neutrophils, whereas in older mice within 6 to 20 months of age, wound healing was slower when depleted of neutrophils [46]. These neutrophils most likely initiate angiogenesis during the granulation phase, and act in parallel with classical neutrophils to induce important bacterial effects during skin repair. On the other hand, clinical observations have revealed that neutrophils are vital for efficient wound repair, as neutropenic individuals often have difficulty healing wounds [47,48]. However, the function of neutrophil-specific effectors that may contribute to wound healing remains unclear. We can say that the neutrophil depletion results in delayed wound healing in older mice, but not in young mice, suggesting that neutrophils undergo functional changes during aging.

Within 48 to 96 h after injury, macrophages migrate into the wound and become the dominant cell population. They induce many factors that drive the different phases of wound healing. Accordingly, the functions of macrophages are varied and change according to their expression [49]. During the initial stage of wound healing, most of macrophages express a pro-inflammatory phenotype characterized by the production of pro-inflammatory molecules, including reactive oxygen species (ROS), nitric oxide, interleukin-1 (IL-1), IL-6, and tumor necrosis factor-alpha (TNF-α). As the wound healing process continues, the macrophages undergo phenotypic transformation and regulate the expression of factors such as insulin-like growth factor 1 (IGF-1), platelet-derived growth factor (PDGF), transforming growth factor-β (TGF-β), and vascular endothelial growth factor-A (VEGF-A). During the final stage of wound healing, the macrophages play a regulatory role by suppressing inflammation through IL-10 production [50,51,52]. Shortly after the publication of these findings, the participation of immune cells and the main contribution of macrophages in wound healing was discovered to be through the secretion of signaling molecules, such as chemokines, cytokines, and growth factors. These cytokines activate and recruit other cells involved in wound healing, such as macrophages and lymphocytes. Through these numerous and various functions, macrophages influence angiogenesis and matrix synthesis. Table 1 summarizes the agents and cytokines produced by macrophages, and their functions.

### 2.2. Chemokines, Cytokines, and Growth Factors

Wound healing is a complex and dynamic biological process coordinated by different cellular events and instructions from the microenvironment. Interestingly, the use of cell-secreted proteins, known as secretomes, has potential in accelerating wound healing [57]. Normally, the instructions exist in the form of chemokines, cytokines, and growth factors, which together organize the phases of healing. Chemokines are a class of bioactive signaling molecules and key regulators of the wound healing process, where they are identified for their role in leukocyte migration [58]. They can classified into the following four families depending on their structure: CC- (28 members), CXC- (17 members), C- (1–2 members), and CX_3_C-chemokines (1–2 members) [59]. In general, chemokines are involved in all phases of wound healing; however, the composition varies, particularly during the inflammation and proliferation phases, in order to promote angiogenesis (Figure 2). During the inflammation phase, the primary function of chemokines is to recruit inflammatory cells to remove debris, dead cells, and foreign bodies from the wound. In addition, the released pro-angiogenic molecules are responsible for facilitating the proliferation, migration, and differentiation of endothelial cells and keratinocytes, which eventually close the wound [60]. The CC-chemokines present during the initial wounding event are CCL1, CCL2, CCL3, CCL4, CCL5, and CCL7, all of which are able to chemo-attract macrophages [61]. Meanwhile, the CXC chemokines that are found in the wound are CXCL1, CXCL2, CXCL5, CXCL7, CXCL8, and CXCL12, where they directly promote angiogenesis. In the proliferation phase, chemokines play an indirect function whereby they facilitate the recruitment of macrophages that secrete growth factors and cytokines to promote angiogenesis. For example, chemokines CCL2 and CCL3 are highly expressed in the wound during this phase. Previous research studies have shown that the administration of CCL2, CCL21, CXCL12, and a CXCR4 antagonist (inducing the broad-spectrum inhibition of the CC-chemokine class) enhances the process of wound healing [60,62].

Furthermore, the skin repair process begins with the release of various growth factors from platelets and immune and surrounding cells. Growth factors are biologically active polypeptides that have a large array of functions (Table 2). Adding growth factors to wound healing models, the duration of the healing process was greatly reduced. The complex process of wound healing is mediated by a network of enzymes that are partially controlled by growth factors. They control the differentiation, growth, and metabolism of the cell involved. The growth factors operative in tissue repair and wound regeneration processes include epidermal growth factor (EGF), PDGF, TGF-α, and TGF-β. Previous studies have reported the useful impacts of growth factors, and all have been shown to promote wound healing [63,64,65]. Another study also reported that FGF can protect cells against apoptosis, as well as induce cellular migration and differentiation [66]. In human nonhealing wounds, the levels of PDGF were found to be decreased compared to those observed in acute surgical wounds [67]. The same observation has been determined through an experimental model with diabetic mice, in which bFGF and VEGF were reduced. Moreover, the diabetic patients also demonstrated lower levels of IGF-1 and TGF-β1 [68]. In recent years, more drug delivery systems have been developed to control the release of growth factors. This can be achieved through the use of smart biomaterial-based dressings. These wound scaffolds are equipped with special features, physicochemical characteristics, and biological properties so that they can provide an effective and safe platform for wound treatment, thus providing numerous benefits to the patient.

## 3. Natural-Based Products for Wound Healing

Even though most acute wounds can heal by themselves, a more proficient and active treatment is needed for nonhealing wounds. Different kinds of treatment options (including both medical and surgical options) are currently available that help in wound repair. Despite their advantages and uses, they face a lot of limitations arising from the delivery system, such as low efficacy, short residence time, high costs, high toxicity, and high risk of infection. This is owing to the fact that chronic wounds can remain unresponsive to conventional wound care treatments, for example, wound dressings, topical agents, and skin grafts. These shortcomings require further exploration into the role of natural-based products, especially biomaterials and bioscaffolds, for treating nonhealing chronic wounds, which potentially could provide reliable solutions in the near future.

In general, biomaterials can be either natural or synthetic, and act as substitutes for biological tissues in the skin layer. Natural-based products and biomaterials can be used directly as medicaments for alleviating the wound or as drug carriers for other therapeutics deliveries [81]. On this basis, scientists have reviewed functional nanomaterial-based dressings, such as hydrogel, gauzes, and hybrid structures, to evaluate the wound state when applying smart wound dressings. Researchers have explored the translation of nanomaterial-based wound dressings and related medical aspects into real-world use [82]. The global growth rate of the biomaterials market size is approximately 15.9%, and it is expected to reach USD 348.4 billion by the year 2027 [83]. Natural products have gained huge popularity as a source of new bioactive ingredients for drug development, leading to rapid growth in biomedical research. As natural biomaterials, collagen [84,85], gelatin [86], chitosan [87], hyaluronic acid [88,89], and alginates [90] have shown promising results for skin wound healing. Recently, Gaspar-Pintiliescu et al. presented the main characteristics and properties of natural biomaterials, the advantages and disadvantages of commercial wound dressings, and the mechanisms involved in wound healing [91]. For example, extensive chemical functionalization using peptides, such as arginine–glycine–aspartic acid (RGD), could stimulate cell adhesion in more than one type of cell. This peptide is able to present the structural characteristics of many proteins in living organisms, and also plays a role in controlling cell differentiation, growth, and behavior [92]. Here, the detailed roles of naturally derived products as biomaterials in wound healing studies, along with their uses, applications, mechanism of action, and outcomes demonstrated in the literature, are presented in Table 3.

From the details demonstrated above, naturally derived therapeutics with topical application is appealing as it provides a local effect while limiting systemic side effects; however, such applications are inhibited by the proteolytic wound environment, which reduces the bioavailability of the drug. The presence of different chemicals in the fabrication process can limit the ability of the researchers to conclude the specific action of the individual chemical and its mechanism of action. A study conducted by Raja revealed that natural substances have a synergistic effect in wound healing and skin regeneration [114]. However, side effects, including irritation and allergic hypersensitivities, were noted. A recent study showed that a collagen type I bioscaffold, derived naturally from ovine tendon, can be implanted into patients within 6–8 h of biopsy. It considered to be very safe; the authors reported no toxic effect on cells, as it promoted higher cell attachment and the proliferation of both primary human epidermal keratinocytes (HEK) and human dermal fibroblasts (HDF), thus, it did not cause any complication systematically [84]. Furthermore, natural-based products or materials may be at high risk of contamination by infectious agents. Hence, proper sterilization and timely microbial testing is necessary before their use. Along this route, collaborative research incorporating nanoparticles, such as silver, will provide a better understanding of how to reduce the limitation of natural-based materials mentioned above. Therefore, this combined approach can be further integrated into wound healing. Moreover, some of our group studies have proved that incorporating nanoparticles within natural-based composites can enhance scaffolding performance, cellular interactions, and their physico-chemical and biological interactions [115]. Our recent report described the efficiency of fabricated bilayer scaffolds, composed of a collagen sponge (bottom layer) and gelatin/cellulose (outer layer) incorporated with graphene oxide and silver nanoparticles, at preventing possible external infections post-implantation [116]. There are many ongoing studies in the literature on integrating antibacterial compounds into natural-based biomaterials with the aim of producing a synergistic effect in skin tissue wound healing, especially for chronic wounds [117].

### 3.1. Clinical Trial

Presently, the clinical application of biomaterials in wound treatment has been in the form of wound dressings, which are able to maintain a moist environment and protect the wound bed [34,118]. Biomaterial research has grown progressively, and seeks to use these dressings to actively encourage wound healing through immune modulation, cell infiltration, ECM generation, and vascularization [119]. A small number of clinical studies have supported the therapeutical potential of using natural-based biomaterial products in human wounds. As previously mentioned, natural-derived biomaterials have shown promise in their use as biological wound dressings due to their inherent properties, including biocompatibility and hemostatic control, and their ability to be modified or functionalized to incorporate into drugs in order to create a bioactive dressing. However, according to Schneider et al. [120], the main limitations of some natural-based polymer biomaterials are their immunogenicity and potential to inhibit cell function in the long term, which results in their degradation, as they are not easily controlled. Among naturally derived wound dressings, many researchers have successfully undertaken clinical studies for wound healing, as summarized in Table 4. Indeed, most of the products were reported to form highly absorptive dressings, highlighting their proved effectiveness in wound management.

From our research findings, the largest clinical trial for wound healing was conducted using Dermagraft, performed by Harding et al. [128], investigating a group of 366 patients with venous leg ulcers (VLU). Their results demonstrate that patients with chronic diabetic foot ulcers of more than 6-week duration experienced a significant clinical benefit when treated with Dermagraft versus patients treated with conventional therapy alone [129,130]. Furthermore, individuals with diabetes mellitus are at an increased risk of developing a diabetic foot ulcer (DFU). With regard to the treatment for nonhealing DFU, Integra dermal regeneration represents an advanced, acellular, and bioengineered matrix that successfully achieved its primary function in a randomized and controlled trial [131]. Following this, Integra treatment reduced the time to complete wound closure, increased wound closure rate, improved quality of life components, and had fewer adverse effects compared with the standard care treatment. The skin replacement layer, which consists of collagen and chondroitin-6-sulfate, has been shown to promote skin regeneration and vascularization in previous clinical studies [132,133].

When a wound enters the chronic phase, there will be the presence of persistent infections. Nonetheless, antibiotics applied topically (owing to superinfection, high level of antibiotic resistance, impaired healing, and delayed allergic reaction) are not recommended unless critical bacterial colonization is recognized [134]. Since the incorporation of antibiotics into the biomaterial structure is not plausible, there is a growing trend in biomaterial engineering for the fabrication of artificial skin grafts and wound dressing products made from natural-based biomaterial matrices reinforced with drug particles (antibacterial and antimicrobial). Kingsley et al. have completed their clinical study evaluating the outcomes of patients with acute bacterial skin and skin structure infection treated with a variety of drugs, including vancomycin, delafloxacin, and linezolid, [135]. There were no significant differences in bacterial eradication among the treatment groups. This case series applied a cost-effective and clinically efficient method of treatment. Glat et al. [136] also presented a study comparing SilvaSorb (an alginate dressing combining with ionic silver technology) with Silvadene (a silver sulfadiazine cream), demonstrating the efficacy of SilvaSorb in the treatment of partial-thickness burns. Furthermore, Simcock and May recently conducted a study on split skin graft reconstruction of scalp defects using a decellularized extracellular matrix biomaterial, with the use of SilvaSorb after application to successfully stop infection at the reconstruction site [137]. The integration of various antibacterial agents into dressings has been clinically tested.

Excitingly, the current treatments that aim to accelerate wound healing that are in the pipeline for clinical trials have taken these concerns into consideration to develop new technologies or concepts for drug delivery. Research has continued in this field, focusing on the development of more advanced wound dressings involving the combination of new extracted natural-based biomaterials to produce synergistic treatment results. For example, Fauzi et al. fabricated collagen sponge from ovine tendon to act as an implant product for wound healing [138,139,140]. The authors used the green Halal source collagen type I (animal waste products from slaughterhouse), which there are more than 80% collagen protein from total protein, via a low-chemical-based method (due to the usage of low acetic acid aqueous). The sponge-like material exhibited a highly porous structure (60–70%) and a proper water vapor transmission rate (~1100 g/m^2^ h^−1^) for optimal wound healing. This product is nontoxic, and showed no immune response through in vitro and in vivo evaluation, as well as in a pre-clinical model for efficiency [141,142]. Considering these characteristics of newly developed materials, the authors conclude that these materials show promise in the management of burn wounds with moderate to high exudate.

### 3.2. Criteria of Randomized Clinical Trials in Wound Care

As the care for acute and chronic wounds becomes a major problem worldwide, many products have been released for wound healing. However, studies evaluating the safety and efficacy of wound care products are frequently limited. Thus, randomized clinical trials are universally recognized where the study design of choice is to compare treatment effects. The use of randomized clinical trials to advance investigations into the effectiveness of interventions seems realistic and advantageous. In the clinical research scenario, there are several criteria that are essentially considered, as summarized in Table 5. Patients for whom the intervention is intended are to be determined via settings from which eligible patients will be selected. Keep in mind the narrow inclusion criteria, which should present a stronger treatment effect, leading to further difficulties in patient recruitment and the generalization of results. Meanwhile, eligible patients should be fully informed about treatment options and, if they decide to participate in a trial, they should provide written informed consent. The consent process must respect the patient’s ability to make decisions and adhere to individual hospital rules for clinical studies [143].

Generally, safety will come first. To set up wound dressing products on the market, early studies will begin at the concept phase and design stages of the products. The main measurements and outcomes concern the reduction in wound size, as well as clinical infection compared to the start of treatment. In addition, the main parameter of the patients is well-being, and treatment-associated adverse events are also focused on [145]. Following the type of technology, there will be a transition to animal studies and, in turn, to clinical trials on humans. The crucial part is the last stage, where engaging in human clinical studies requires the work of clinicians, research and development staff, statisticians, and others to agree on the study designs.

## 4. Summary and Outlook

Over the past few decades, the advancement and improved understanding in material science, bioengineering, and medicine has led to great achievements in wound healing. The use of naturally derived materials for the wound healing is one of the fundamental principles in skin tissue engineering. It can be referred to the unique properties and ability of natural materials to interact with different bioscaffolds. The transition from 2D cell culture to 3D bioscaffolding has evolved as more and more studies have created significant changes in morphology, cell migration, differentiation, and viability [146]. Consequently, understanding of critical cellular functions throughout the wound healing processes and progression of wounds is vital. Continuing this theme, there is great potential for the development of sustainable, safe, nontoxic, and effective materials extracted from plants, animals, or other natural sources [147]. Considering the different wound types and advancement in the regenerative medicine, this paper describes the understanding of the immune cell concepts in wound healing, the growing trend towards the natural-based biomaterial scaffolds developed in recent years, and taking into account the clinical trials in the domain.

In wound healing, a wound dressing might not be sufficient to ensure proper healing [148]. Hence, biomaterial scaffolds are often implied. Ideally, bioscaffolds are highly biocompatible and maintainable in skin tissues, thus providing shape, mechanical support, and appropriate microarchitecture for cellular growth and reorganization to stimulate the recovery process [149,150,151]. Similarly, biomaterials used for wound healing should destroy and/or repel microbes and other infectious agents, be hydrophilic and porous enough to absorb exuding fluids, and/or have a swelling factor large enough to fill any voids in the damaged tissue. Depending on the goal of treatment, bioscaffolds can be designed to control moisture content in wounds, prevent infection, and/or maintain an optimal microenvironment, including temperature and pH. Meanwhile, biological properties must be considered, such as hydrophilic properties (the scaffold can be either hydrophilic or hydrophobic to control the rate of the liquid passage from the wound), the ratio of porosity and swelling (to allow the encapsulated drug to penetrate the wound), and degradation (to release biomaterial into the wound and help tissue regeneration). On the other hand, tissue grafts are one of the most used naturally derived scaffolds. Due to their resemblance to native tissues, along with their ability to promote cell attachment, proliferation, and organization of the cells, tissue grafts have been demonstrated as the most convenient and effective implantable devices [152].

In wound care, a clinical trial is an investigation that uses human volunteers to investigate the efficacy and safety of a new medical product. Nonetheless, trials can be conducted when only practical laboratory and animal studies have been undertaken. Investigating the wound healing processes that enable wounds to heal is primarily carried out using models involving in vitro (cells), animal, and human data. In general, it is accepted that the use of human models presents the best chance and opportunity to understand the factors that influence wound healing, as well as to evaluate the effectiveness of treatments applied to wounds [153].

## 5. Challenges and Limitations for Clinical Trial Wound Healing

In the development of natural-based biomaterials for wound healing, there are many challenges to consider. First is the parameters required for treatment, including the ability to withstand the proteolytic wound environment to ensure the bioavailability of the active agents. Thus, the incorporation of drugs with scaffolds or co-administration by implanted products will prolong the release. Studying the effects of biomaterial treatment or implantation on wounds (acute, burns, chronic) is more difficult. Lesions are less common, and the right patient is harder to find. Additionally, many researchers overestimate the number of lesions they can generate. Because of the nature of the disease (the wound type) and its treatment (surgery, dressings), it is also highly problematic to conduct blinded studies. Furthermore, the treatments need to be cost-efficient to penetrate the market; when trial implementation is included in a business plan, the cost and complexity of virtually any type of trial is often underestimated. Today, most newly developed biomaterial products are very expensive, which limits clinical availability. To achieve the successful development of biomaterials as viable clinical options, special consideration must be given in choosing the best-fitted dressing for a specific wound and patient’s primary diseases. Hence, wound management strategies should progress towards minimizing costs while maintaining optimal clinical decision [154]. It was suggested that better collaboration should be encouraged between industry market segments, clinical research, and clinical practice to achieve such a perspective.

## 6. Conclusions

To summarize, wound supervision continues to be a topic of high interest in the tissue engineering and regenerative medicine field, aimed at developing better biomaterial selection for a variety of dressings and scaffolds. There is a broad range of available products, including hydrogels, foams, sponge, films, and other biomaterial scaffolds, that have been reported in the literature with different points of success in wound healing. Meanwhile, some of the products have already entered the market, and are currently used in clinical practice. The use of naturally derived substances is an exciting, clean, safe, and brilliant innovation for wound treatment. The development of these scaffolds requires solutions to the clinical and medical challenges faced in the treatment of nonhealing chronic wounds. Natural-based compounds generally exhibit promising properties, such as antioxidant, anti-inflammatory, angiogenic, and cell synthesis-modulating components, which are crucial biological functions necessary for wound healing. The multiple advantages over synthetic biomaterials supports the use of natural components in wound healing research. Nevertheless, more clinical trials should be carried out to deliver concrete evidence to support the utilization of naturally derived biomaterials in the management of wound healing. Moreover, more research is needed to understand the mechanisms of action behind their therapeutic effects.

## Figures and Tables

**Figure 1 biomedicines-10-02226-f001:**
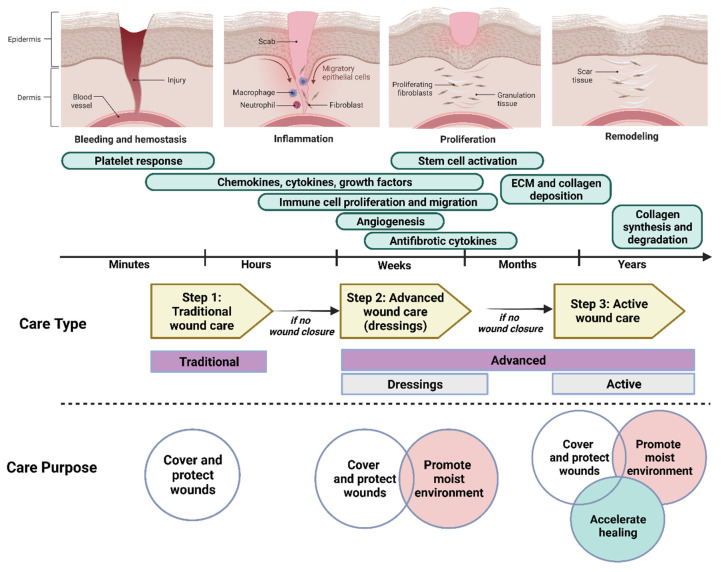
Four different phases of wound healing that are involved different cellular events and mechanisms, as well as the type and purpose of wound care used at different time points after injury. Created with BioRender.com (accessed on 26 July 2022).

**Figure 2 biomedicines-10-02226-f002:**
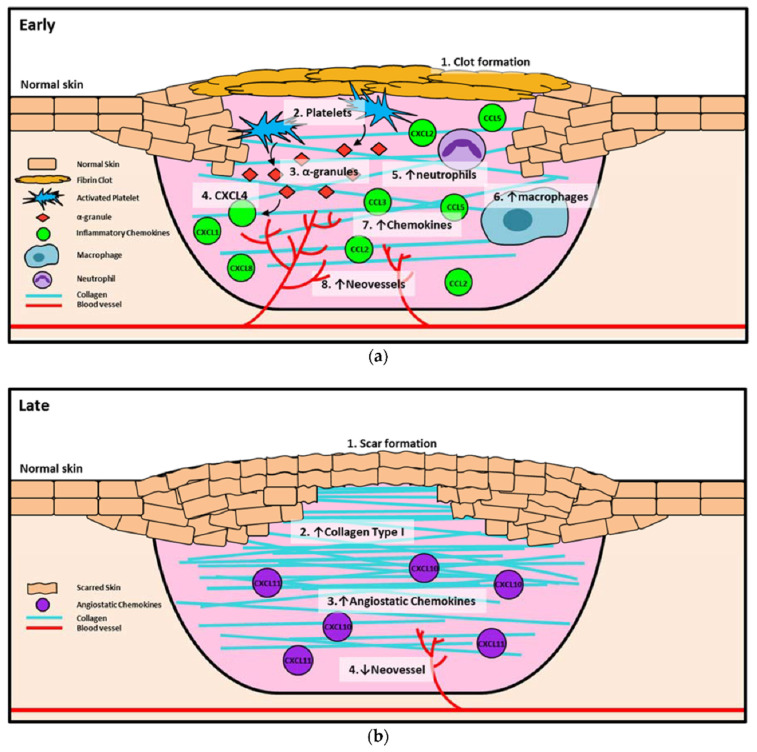
Chemokines in early and late phases of wound healing. (**a**) Early wound healing, including clot formation, inflammation, and proliferation. (1) Clot formation occurs to prevent the loss of blood, and (2) platelets are activated and release (3) α-granules, which in turn release (4) CXCL4 as an early inhibitor of angiogenesis. Once the clot has fully formed, other chemokines such as CXCL8, CXCL1, and CXCL2 are released by α-granules to recruit inflammatory cells, including (5) neutrophils and (6) macrophages. Neutrophils are increased early in the healing process, then macrophages soon take over as the primary inflammatory cell. Neutrophils and macrophages release (7) chemokines such as CCL2, CCL3, and CCL5 into the wound to promote the recruitment of more inflammatory cells that release pro-angiogenic growth factors, which in turn (8) increase neovessel formation in the wound. (**b**) Late wound healing is the remodeling stage. In this stage, the wound is fully healed and (1) a scar has formed. Type III collagen converts to (2) type I collagen to promote scar formation and create a more stable wound seal. During the remodeling process (3), angiostatic chemokines (CXCL10, CXCL11) promote the (4) regression of neovessels, as there is no longer a requirement for enhanced blood flow or the recruitment of immunological cells to the site. The symbol ↓ indicates decrease; ↑ indicates increase. The figure has been reprinted (adapted) with permission from Ref. [60] under the terms and conditions of the Creative Commons Attribution (CC BY) license (http://creativecommons.org/licenses/by/4.0/ (accessed on 26 June 2022)).

**Table 1 biomedicines-10-02226-t001:** Cytokines secreted by macrophages, and their functions. The symbol ↑ represents increase, while ↓ represents decrease. The details are adapted from [42,53,54,55,56].

Cytokine	Endothelial Cell Proliferation	Angiogenesis	Fibroblast Proliferation	Collagen Synthesis
TNF-α	↑↓	↑	↑↓	↑↓
IL-1	↓	↑↓	↑↓	↑↓
Il-6	↓	↑	↓	↑
TGF-β	↑↓	↑	↑↓	↑
TGF-α	↑↓	↑	↑↓	↑↓
PDGF	↑	↑	↑	↑
IGF-1	↑	↑	↑	↑

**Table 2 biomedicines-10-02226-t002:** Summary of biological properties and respective roles of the major growth factors that participate in wound healing.

Growth Factor	Biological Activities	Functions	Reference
PDGF	Regulate synthesis of matrix components Increase proliferation of fibroblasts	Inflammation Granulation tissue formation Re-epithelialization Matrix formation and remodeling	[69,70,71]
EGF	Increase proliferation of keratinocytes and endothelial cells Increase EGF binding to EGF-R Increase production of IGF	Re-epithelialization	[72,73]
IGF	Induce proliferation of keratinocytes Decrease protein catabolism (fibroblasts)	Inflammation Re-epithelialization	[74,75,76]
TGF	Regulate cell proliferation and matrix component synthesis Decrease growth of fibroblasts and keratinocytes Increase expression of keratin Increase proliferation of fibroblasts	Inflammation Granulation tissue formation Re-epithelialization Matrix formation and remodeling	[77,78]
FGF	Synthesis and deposition of various ECM components Increase keratinocyte motility during re-epithelialization	Granulation tissue formation Re-epithelialization Matrix formation and remodeling	[79,80]

**Table 3 biomedicines-10-02226-t003:** Comprehensive details on the role of naturally derived products in wound healing studies.

Naturally Derived Product	Sources	Type of Formulation	Uses and Applications in Wound Healing	Possible Mechanism of Action	Wound Model Used or Type of Study	Outcome	Reference
Gelatin	Bovine skin	Topical gel	Care for acute wounds	Lowering the oxidative damage and increase in the production of collagen	In vivo (mice)	Keep the wound area clean, warm, and moist. Enhance wound healing by reducing the wound size	[93]
Collagen	Bovine Achilles tendon, sheep ovine tendon	Topical	Treatment of full thickness wounds	Control the bacterial growth in the wound environment	In vivo (rats)	Faster wound healing process with high recovery percentage (wound healing rate)	[94,95,96]
Hyaluronic acid	Polysaccharides	Topical gel	Treat chronic ulcers	Anti-inflammatory effects	Clinical studies	Stronger regenerative potential in epidermal proliferation and dermal renewal	[97,98]
Chitosan	Shells of crustaceans	Topical	Treat diabetic wound	Present hemostatic action, which can be exploited to enhance healing	In vivo (rats)	Promotes tissue regeneration with improved function	[99,100,101]
Alginate	Kelp-like Phaeophyceae	Topical	Treat chronic and diabetic wounds	Maintain a physiologically moist environment and minimize bacterial infections at the wound site	In vivo (mice)	Reduces healing time and wound size	[102,103,104]
Elastin	Bovine neck ligament	Topical gel	Wound repair and dermal regeneration	Recruit and modulate macrophages to facilitate tissue regeneration	In vivo (mice)	Promotes innate immune cells, angiogenesis, and collagen regeneration	[105]
Silk fibroin protein	*Bombyx mori*, the domestic silk moth	Topical	Skin repair and wound regeneration	Conducive microenvironment for wound healing (excellent fluid handling, air-permeable, and bacterial barrier properties)	In vivo (rabbit and porcine) and clinical trial	Promote wound healing speed. Prior to the clinical trial, wounds treated with the silk fibroin healed ~14 days post-surgery, which was remarkably faster than the untreated control (21 days)	[106]
Carrageenan	Seaweeds	Topical gel	To treat full-thickness wounds	Strong antibacterial activity to destroy *Staphylococcus* epidermis and *Escherichia coli* within 3 h of incubation	In vivo (rats)	Wound area reduction. Excellent wound healing effect (1.3% wound area after 2 weeks)	[107]
Aloe vera	Not specified	Topical gel	Treat various ailments of the skin due to its anti-inflammatory and antimicrobial properties	Stimulate the release of several growth factors	In vivo (rats)	Increase in rate of contraction of wound area	[108,109,110]
Honey	Not specified	Topical	Antioxidant, antimicrobial, and anti-inflammatory properties	Wound healing effects are due to its antibacterial action, high acidity, osmotic effect, antioxidant, and hydrogen peroxide content	Clinical studies	Honey was not found to benefit chronic venous leg ulcers; lack of statistical evidence to prove the use of honey on superficial and partial thickness burn wounds	[111,112]
Cocoa	Not specified	Topical	Treat various ailments of the skin	Improves re-epithelialization	Porcine model	Wound healing improved, but limited studies have claimed the above results	[113]

**Table 4 biomedicines-10-02226-t004:** Some examples of natural-based biomaterial dressings, demonstrating their effects in the clinical usage for wound healing.

Type	Constituent	Indications	Description	Examples
Hydrogels [121]	Alginate 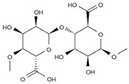	Heavily exuding ulcer, hemorrhagic ulcer, necrotic wounds, dry chronic wounds, and burn wounds	High-level water content (70–90%) in the dressing absorbs minimal fluid but contributes moisture to the wound. Soft elastic properties allow removal without damaging tissue. Some hydrogels lower the temperature of the wound thus providing a soothing and cooling effect	Maxorb, Calcicare, Seasorb, Sorbsan, Nuderm
Foams [122]	Collagen 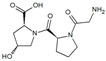	Traumatic injury, hemorrhagic ulcers	A highly absorbent dressing with a hydrophilic surface to interact with the wound and a hydrophobic surface to the environment. Protect against infection and dehydration of wounds. Can be left on the wound for several days	Promogran, Puracoll, Prisma, Fibracol, Cellerate
Hydrofibers [123]	Cellulose 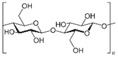	Heavily exuding ulcers and infected wounds	Highly absorbent	Tegaderm matrix, Prisma, Silvercel, Aquacel, Promogram, Dermafill Xylinum Cellulose
Wafers/hydrocolloids [124,125]	Carboxymethylcellulose 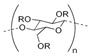	Mildly exuding ulcers	Two-layer dressing: inner layer has moderate absorbency, and the outer waterproof to protect against bacteria. The hypoxic environment created aids autolytic debridement	DuoFilm, Exuderm, DuoDERM, RepliCare
Films [126,127]	Polyurethane (natural oil polyols) 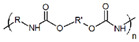 Pectin 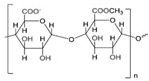 Starch 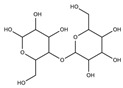	Superficial wounds with low exudate, Minor burns, pressure areas, donor sites, postoperative wounds, and various minor injuries including abrasions and lacerations	A thin flexible sheet of nonabsorbent transparent adhesiveprotects against bacteria and fluids, and the autolyticnature of debridement	Blisterfilm, Comfeel film, ClearSite, Procyte, OpSite

**Table 5 biomedicines-10-02226-t005:** Checklist of criteria to be defined and completed for an optimum design in wound care clinical trials [144].

Criteria	Descriptions
Setting	The trial setting (e.g., home care, general hospital, nursing home, or specialized (university) clinic) is defined
Patients	Eligibility criteria for patients are described (inclusion and exclusion criteria) Written informed consent will be obtained from every patient included
Interventions	The treatment to be used in each trial arm is standardized Cointerventions are allowed but prespecified (the same in both trial groups)
Outcomes	Primary and secondary outcomes are predetermined It is described when and how outcomes are evaluated
Sample size	Sample size is calculated (calculation based on the expected clinically relevant difference in primary endpoint)
Randomization	The unit of randomization is defined (e.g., the wound or the patient) The allocation sequence is randomly generated The treatment allocation is adequately concealed
Blinding	It is defined who is blinded after assignment to the intervention and how, including: - Patients (recommended) - Caregivers (recommended) - Outcome assessors (strongly recommended)
Intention-to-treat	All randomized patients are to be analyzed in the group to which they were allocated
Funding	Funding through unrestricted grants only
Follow-up	Duration of follow-up is defined
Ethics	Ethics review board approval Trial registration

## Data Availability

The data presented in this study are available on request from the corresponding author.

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
