# Peer review of "The Discovery and Development of Natural-Based Biomaterials with Demonstrated Wound Healing Properties: A Reliable Approach in Clinical Trials"

_biomedicines, 2022, doi:10.3390/biomedicines10092226_

Round 1

Reviewer 1 Report

The manuscript gives a review of the natural-based biomaterials with wound healing properties. However, this review is not well organized and the chapter after 3.1 is missed. The importance and development of natural biomaterials for wound healing are not fully clarified. I thus recommend rejection.

Author Response

Please refer to pdf attachement.

Reviewer 2 Report

Very well written basic review of wound healing with emphasis on existing materials or biological origin.  This paper is an effective overview, could be strengthened by more detailed biochemistry and processing of the materials described

Author Response

Please refer to pdf attachment.

Reviewer 3 Report

This review summarizes aclual and novel information about natural-based biomaterials with wound healing properties. The review can be accepted for publication after minor revision. The following issues should be clarified.  

Please add information about care type and care purpose from Fig.1. What does mean active wound care? Maybe is a sense to add information about "smart" or responsive systems as the most advanced technology.

The information about the results presented in Table 4 is absent in the main text.

Information about polyurethane (natural oil polyols) is absent in reference 115.

What is the chemical structure of the hydrocolloid from Table 4, references 113, 114? In other cases, the chemical structures of the components were pointed out.

I am not sure but maybe RGD-based natural materials are important to mention in this Review. 

I think it is important to cite the following references where prospective materials based on the natural components were developed:

 https://doi.org/10.1016/j.colsurfb.2014.03.049

https://doi.org/10.3390/ma13020337

Author Response

Please refer to pdf attachment.

Reviewer 4 Report

This review article discusses the latest discoveries on natural-based wound healing implanted products. The wound healing mechanism, the existing challenges, and future perspectives on the clinical trial of natural-based dressing materials are reviewed. The following comments should be addressed before the manuscript can be considered further:

There are many articles about natural-based wound dressing materials (e.g., 10.1021/acsabm.2c00035; https://doi.org/10.1021/acs.biomac.0c0076010.1021/acsnano.1c08411; 10.1021/acsnano.1c04206; https://doi.org/10.1016/j.ijbiomac.2019.07.155). The authors did not mention these existing reviews and justified the importance/novelty of the present review.

Authors should provide more details on how to perform clinical trials of wound dressings. The current version of the manuscript is devoted mainly to the wound healing mechanism and does not reflect the main purpose of the manuscript.

The authors should revise Table 3 and include more examples of natural polymers and proteins used to develop wound dressing materials. Additionally, the authors stated in Table 3 that the gelatin source is soy lecithin, which is incorrect.

After reading the title of the manuscript, I do not think it reflects its entire content. Neither natural-based wound dressings nor clinical trials of such materials are essentially covered. Hence, I suggest a major revision of the manuscript for further consideration for publication.

Author Response

Please refer to pdf attachment.

Round 2

Reviewer 1 Report

Revisions addressed satisfactorily.

Author Response

As stated in pdf doc.

Reviewer 4 Report

Although an appreciable effort was made to improve the manuscript, the authors did not demonstrate the main novelty of the work. In this regard, the authors should discuss this information in the introduction section.

Author Response

As stated in pdf doc.
